# Naphthalocyanine-Based NIR Organic Photodiode: Understanding the Role of Different Types of Fullerenes

**DOI:** 10.3390/mi12111383

**Published:** 2021-11-11

**Authors:** Siti Anis Khairani Alwi, Syaza Nafisah Hisamuddin, Shahino Mah Abdullah, Afiq Anuar, Atiqah Husna Abdul Rahim, Siti Rohana Majid, Tahani M. Bawazeer, Mohammad S. Alsoufi, Nourah Alsenany, Azzuliani Supangat

**Affiliations:** 1Low Dimensional Materials Research Centre, Department of Physics, Faculty of Science, Universiti Malaya, Kuala Lumpur 50603, Malaysia; sitikhairani96@gmail.com (S.A.K.A.); syazanafisah09@gmail.com (S.N.H.); shahinomah@gmail.com (S.M.A.); afiqanuar44@gmail.com (A.A.); 2Centre for Ionics University of Malaya, Department of Physics, Faculty of Science, Universiti Malaya, Kuala Lumpur 50603, Malaysia; husnatiqah@gmail.com (A.H.A.R.); shana@um.edu.my (S.R.M.); 3Department of Chemistry, Faculty of Applied Science, Umm Al-Qura University, Makkah 24382, Saudi Arabia; tmbawazeer@uqu.edu.sa; 4Mechanical Engineering Department, College of Engineering and Islamic Architecture, Umm Al-Qura University, Makkah 24382, Saudi Arabia; mssoufi@uqu.edu.sa; 5Department of Physics, Faculty of Science, King Abdulaziz University, Jeddah 21589, Saudi Arabia; nsenany@kau.edu.sa

**Keywords:** organic photodiode, NIR, naphthalocyanine, fullerene, solution-processed

## Abstract

In this work, we presented experimental observation on solution-processed bulk heterojunction organic photodiode using vanadyl 2,11,20,29-tetra tert-butyl 2,3 naphthalocyanine (VTTBNc) as a p-type material. VTTBNc is blended with two different acceptors, which are PC_61_BM and PC_71_BM, to offer further understanding in evaluating the performance in organic photodiode (OPD). The blend film of VTTBNc:PC_71_BM with a volumetric ratio of 1:1 exhibits optimized performance in the VTTBNc blend structure with 2.31 × 10^9^ Jones detectivity and 26.11 mA/W responsivity at a −1 V bias. The response and recovery time of VTTBNc:PC_71_BM were recorded as 241 ms and 310 ms, respectively. The light absorption measurement demonstrated that VTTBNc could extend the light absorption to the near-infrared (NIR) region. The detail of the enhancement of the performance by adding VTTBNc to the blend was further explained in the discussion section.

## 1. Introduction

Organic photodiodes (OPDs) are growing tremendously as an alternative to inorganic photodiodes due to their unique advantages of large-area fabrication, lightweight, low cost, detection tunability and mechanical flexibility [1,2,3,4,5]. For the last few years, the solution-processed bulk heterojunction (BHJ) was the prevalent design used in OPDs and organic photovoltaics (OPVs) [6,7,8,9]. It appears to be the most effective approach for an organic semiconductor with respect to the processing costs. This design mixes the interfacial region between donor and acceptor on a few nanometres scales to enhance charge separation. The progress in OPDs development contributes to the production of many applications such as image sensors, cameras, remote sensing, camcorders, automation and optical transmission system [6,10]. Therefore, the key parameters of an OPD such as detectivity, responsivity and external quantum efficiency (EQE) should be enhanced to achieve a high OPD performance.

In the fabrication of BHJ OPDs, the commonly used electron acceptor material is based on fullerene derivatives. Among others, [6, 6]phenyl-C61-butyric acid methyl ester (PC_61_BM) and [6, 6]phenyl-C71-butyric acid methyl ester (PC_71_BM) are the most widely used since they possess high electron affinities and high solubility in various organic solvents [6,8,11]. The combination of matching energy level between donor and acceptor, material miscibility and charge transport properties are considered to be the key factors in the success of this system [12]. Therefore, a clear understanding of donor and acceptor characteristics is needed to obtain high-performance OPDs. In order to achieve high-performance OPD at the near-infrared (NIR) region, a low bandgap small molecule with absorption tendency at the longer wavelength can be used. It is reported that naphthalocyanine molecules with different side groups possessed a promising potential as a donor in OPDs for NIR region detection [13,14]. Naphthalocyanine is a derivative of phthalocyanine that has larger polarizability and is sensitive to the local electrostatic environment [15]. It also contains a more π-conjugated electron system compared to phthalocyanine. The extended π-conjugated electron system of naphthalocyanine led to the smaller band gap and contributed to the red-shifting behavior in the NIR absorption region [16]. This potentially gives naphthalocyanine an advantage to be efficiently used in more NIR OPD applications.

Herein, we investigated the OPD performance using vanadyl 2,11,20,29-tetra tert-butyl 2,3 naphthalocyanine (VTTBNc) as a donor material with fullerene as the acceptor in OPDs BHJ structure. The use of VTTBNc as a donor in the structure is due to the ability of this material to absorb light in the NIR region and its low band gap property. Generally, the low band gap of donor offers a smaller energy offset between highest occupied molecular orbital (HOMO) or lowest unoccupied molecular orbital (LUMO) of donor and acceptor and consequently minimizes the loss due to the charge transfer state between donor–acceptor [17]. In this study, PC_61_BM and PC_71_BM were used as an acceptor to extend the understanding of OPD performance behavior. The utilization of the VTTBNc:PC_61_BM and VTTBNc:PC_71_BM blended materials as the photoactive layer was made as the materials provide a complementary absorption profile in visible to the NIR region. Prior to that, a single material OPD based on PC_61_BM and PC_71_BM was also fabricated as a reference.

Besides, both naphthalocyanine and fullerene are categorized in the small molecule group. Small molecules are organic compounds that possess defined molecular structures with low molecular weight. They are free from batch-to-batch variation and consequently improve the fabrication repeatability compared to the conjugated polymer. Dong et al. emphasized that the improvement of fabrication repeatability can lead to a greater tendency to form ordered domains and provide higher charge carrier mobilities [18]. On top of that, small molecules evade unwanted features of macromolecules such as chain twists and chain-end defects that give rise to structural disorder and low-lying trap states [19]. It is also much cheaper than conjugated polymers and thus making it a reasonable material to develop and study.

## 2. Experimental

VTTBNc and PC_61_BM were purchased from Sigma Aldrich (St. Louis, MO, USA), while PC_71_BM was obtained from Luminescence Technology Corp. (Hsinchu, Taiwan). The molecular structure of VTTBNc, PC_61_BM and PC_71_BM are illustrated in Figure 1a–c. In the preparation of the photoactive solution, 20 mg of each material were dissolved separately in 1 mL chloroform and stirred overnight in the nitrogen-filled glove box. Then, the blend solution was prepared by mixing VTTBNc with PC_61_BM and PC_71_BM in three different volume ratios (1:0.5, 1:1, 1:1.5) and filtered using a 0.20 μm nylon filter. Meanwhile, the pre-patterned indium tin oxide (ITO) coated glass substrate (Ossila, Sheffield, UK) was sequentially cleaned in an ultrasonic bath with soap water, deionized (DI) water, acetone, isopropanol and deionised (DI) water (Sigma Aldrich) for 15 min each and dried with nitrogen purge before being treated with UV-ozone for another 5 min. Next, a poly(3,4-ethylenedioxythiophene) polystyrene sulfonate solution (PEDOT:PSS, PH1000 from H.C Stack, Newton, MA, USA) was filtered using a 0.45 μm nylon filter (LabServ, Derbyshire, UK) and spin-coated (Laurell, North Wales, PA, USA) at room temperature at 3000 rpm for 60 s on top of cleaned ITO substrate to form a 40 nm thin layer and annealed for 30 min at 130 °C as the hole transport layer. The photoactive layer was formed by spin-coating the blended solutions at 2000 rpm for 30 s on top of the PEDOT:PSS surface and annealed at 110 °C for 20 min. The sample was transferred to the thermal evaporator for the deposition of aluminum top contact under a base pressure of 2 × 10^−7^ torr through a shadow mask that eventually formed a 0.045 cm^2^ photoactive area. Finally, the samples were encapsulated with the glass using epoxy glue (Ossila) and cured under UV light. The schematic diagram of OPD is shown in Figure 1d.

In order to determine the energy bandgap of the VTTBNc, the cyclic voltammetry (CV) technique was used. The CV was set up using platinum as the counter electrode with Ag/AgCl as the reference electrode and ITO as the working electrode. The supporting electrolyte was prepared by using 0.01 M hydroquinone in HCl with pH 3.0 (Sigma Aldrich). In order to support this, the ultraviolet photoelectron spectroscopy (UPS) measurement of the BL3.2U was also conducted at Synchrotron Light Research Institute (SLRI, Nakhon Ratchasima, Thailand). Next, the current–voltage characteristics were obtained using Keithley 236 Source Measure Unit equipped with an Oriel (Xenon arc lamp) (Tektronix, Beaverton, OR, United States) solar simulator with 100 mW/cm^2^ input power illumination, and the external quantum efficiency (EQE) was acquired using PVE300 Photovoltaic EQE (IPCE) system (Bentham, Berkshire, UK). The absorption spectra of the photoactive layers were observed with ultraviolet-visible (UV-Vis) spectroscopy (LAMBDA 900, Perkin Elmer, Waltham, MA, USA). Furthermore, characterizations of photoactive layers were observed using Raman and photoluminescence (PL) spectroscopy under 325nm excitation wavelength. Lastly, the Atomic Force Microscopy (AFM) images of the photoactive layers were also obtained using Hitachi AFM5100 using S1-DF3 cantilever (Hitachi, Tokyo, Japan).

## 3. Results and Discussion

### 3.1. Absorption Characteristics

Figure 2 shows the absorption spectra of the single-component material and the blended films from the 300 nm to 900 nm wavelength region. Figure 2a illustrates the absorption spectra of the VTTBNc, PC_61_BM and PC_71_BM. It was observed that VTTBNc exhibits two broad peaks at B-band (333 nm) and Q-band region (761 nm and 810 nm). These absorption peaks are similar to the absorption spectra reported by Dhanya and Menon (2012) [20]. The absorption peak at the B-band correlates to the excitation of localized π-π* transition that occurred between the bonding and anti-bonding molecules. Meanwhile, the absorption peak at the Q-band is related to the charge transfer within the core unit [20,21]. Next, PC_61_BM reveals a maximum absorption peak located at 335 nm, and PC_71_BM has several shoulder peaks observed at 373 nm and 473 nm. Figure 2a also shows that PC_71_BM has a stronger and broader absorption range in the visible spectrum compared to PC_61_BM. The less symmetrical structure of PC_71_BM helps to relax the forbidden transitions in longer wavelengths and increases light absorption compared to PC_61_BM [22,23].

Figure 2b,c illustrates the absorption spectrum of VTTBNc:PC_61_BM and VTTBNc:PC_71_BM, respectively. Both graphs show that the binary blend of VTTBNc:PC_61_BM and VTTBNcPC_71_BM exhibit a broader light absorption spectrum compared to the light absorption of a single material. The addition of PC_71_BM as an acceptor also reduced the low absorption valley in VTTBNc film between 384 nm and 639 nm, while the addition of PC_61_BM only diminished the low absorption valley between 460 nm and 618 nm. Thus, it can be inferred that blends with PC_71_BM have a stronger absorption in the visible range compared to the blend with PC_61_BM. Additionally, both blends with a blend ratio of 1:0.50 exhibit high absorption at the Q-band region due to high VTTBNc content in the blend. VTTBNc:PC_61_BM shows almost the same peak intensity at the shorter wavelength for all ratios. However, the absorption peak at 328 nm in VTTBNc:PC_71_BM reduces with increasing PC_71_BM content. It is also evident from the two graphs that the absorption peak at 700 nm to 900 nm decreased with the increase in PC_61_BM and PC_71_BM content.

### 3.2. Determination of VTTBNc Energy Bandgap

The optical band gap is commonly determined using the UV-Vis absorption spectrum and the Tauc plot model [24]. The optical band gap energy can be determined by simply extrapolating a line to the linear region of the Tauc plot graph plotted from the absorption spectrum. From the Tauc plot shown in Figure 3a, the Q-band gap energy (E_g1_) and B-band gap (E_g2_) of VTTBNc are measured as 1.36 eV and 2.57 eV, respectively. This result is slightly low compared to the band gap reported by Dhanya and Menon (2012), which is between 1.7 eV and 1.8 eV of the Q-band gap and 3.1 eV and 3.4 eV of the B-band gap [20]. The difference in the band gap value is due to the red-shifted property in the absorption range observed in this experiment, resulting in lower band gap values. This property is good in enhancing the OPDs detection at the longer wavelength. Q-band gap energy (E_g1_) was chosen due to half of the solar photon flux occurring at the low band gap energy and also essential in the development of NIR OPD [25].

Next, the HOMO and LUMO levels of VTTBNc are determined by measuring the oxidation and reduction onset potential using CV measurement [26,27]. A proper alignment of HOMO-LUMO energy levels between donor and acceptor is essential for the electron transfer in the device. The HOMO and LUMO energy levels of VTTBNc is calculated using the equations below:HOMO = −e (E_OX_ + 4.4 eV)(1)
or
LUMO = −e (E_RED_ + 4.4 eV)(2)
where E_OX_ and E_RED_ refer to the oxidation and reduction onset potentials, respectively, and the 4.4 eV refers to the absolute potential of Ag/AgCl electrode relative to the vacuum level [28,29]. From the CV plot in Figure 3b, it is observed that VTTBNc altered the oxidation and reduction peaks when compared with the bare ITO. The oxidation and reduction onset potential of VTTBNc are observed at 0.742 V and −0.662 V, respectively. Thus, the HOMO and LUMO levels of VTTBNc are determined to be at −5.14 eV and −3.74 eV, respectively, with a band gap energy of 1.4 eV. The value of band gap energy is almost the same as the value recorded from UV-Vis (1.36 eV). This suggests a small interface barrier between the electrode and VTTBNc thin film in the CV system that gives rise to the equilibrium state occurring quickly [30,31].

Besides CV, the determination of energy levels can also be obtained using the ultraviolet photoelectron spectroscopy (UPS) technique by determining their HOMO level. From UPS spectra (Figure 4), the HOMO_onset_ and cut-off energy of VTTBNc are observed at 0.70 eV and 35.06 eV, respectively. The HOMO level can be determined by utilizing the following equations:(3)Work function=hv−Cuttoff energy 
(4)$$\mathrm{HOMO}=\mathrm{Work}\;\mathrm{function}+{\mathrm{HOMO}}_{\mathrm{onset}}$$
where *hv* (39.5 eV) is the energy of an incident photon used in this experiment. By using the cut-off energy and the HOMO_onset_ observed in the UPS spectra, the HOMO level is determined to be 5.14 eV. This result supports the HOMO level determined in CV as both methods obtained the same HOMO level value, which is 5.14 eV. Meanwhile, the HOMO and LUMO energy levels of PC_61_BM and PC_71_BM are obtained from the literature [32]. The work function and energy levels alignment for all materials are illustrated in Figure 5.

The low band gap could result in smaller energy offset between HOMO and LUMO of donor and acceptor material. This allows the reduction in open-circuit voltage loss by minimizing the loss due to the charge transfer state between donor and acceptor [17]. Therefore, the utilization of light absorption in the NIR region was expected to enhance the photocurrent generation of the device. It is also suitable to make transparent OPD with high transmittance in the visible regions by utilizing such high NIR absorbing material.

### 3.3. Performance of VTTBNc:PC_61_BM and VTTBNc:PC_71_BM Organic Photodiode

Figure 6 and Figure 7 show the current density–voltage (J–V) graphs of single materials and the BHJ OPDs examined in dark and light conditions. The J–V characteristics are investigated in reversed biased conditions to study the dark current and photocurrent generation, which later determine the responsivity and detectivity of the OPDs. The responsivity of the device refers to the electrical output produced per incident power input and can be calculated using the following equation:(5)$$R=\frac{\left(J_{\mathrm{ph}}-J_{D}\right)}{P}\;\;$$
where *R* is the responsivity, *J*_ph_ is the photocurrent density, *J*_D_ is the dark current density and P is the power of incident light. Meanwhile, detectivity (*D**) is defined as the ability of the OPD to detect a small photon signal which is calculated by the equation below [10,33]:(6)$$D^{*}=\frac{R}{\sqrt{2q.J_{D}}}$$
where q is the elementary charge (1.602 × 10^−19^ C). From the equation above, the shot noise (2q.*J*_D_) from dark current is often assumed to be the dominant contribution that limits the detectivity, while flicker noise and thermal noise are ignored [10,33].

As seen in Figure 6, it is observed that PC_71_BM generated a high photocurrent that leads to higher responsivity compared with PC_61_BM. The responsivity of PC_71_BM is measured as 16.40 mA/W, 20.66 mA/W, 21.20 mA/W at 0 V, −0.5 V, −1 V bias, respectively. While the responsivity of PC_61_BM is measured as 10.50 mA/W, 14.32 mA/W, 16.21 mA/W at 0 V, −0.5 V, −1 V bias, respectively, as listed in Table 1. The higher photocurrent generation in PC_71_BM devices is mainly due to the stronger absorption in the visible region, as observed in Figure 2. However, a high dark current was also observed, causing the device to have lower detectivity compared to PC_61_BM. This may be attributed to the high thermal generation of charge carriers at the contacts when the voltage is applied [33]. As PC_71_BM obtained stronger absorption and higher photocurrent generation, it also contains a higher charge carrier generation compared to PC_61_BM, and in turn, tends to have a high thermally generation of charge carriers from the two electrodes contact.

In order to improve the OPD performance, VTTBNc is blended with PC_61_BM and PC_71_BM in three different ratios. The J–V graphs of VTTBNc:PC_61_BM and VTTBNc:PC_71_BM are shown in Figure 7a,b. For the VTTBNc:PC_61_BM devices, the result shows the increase in the photocurrent density with increasing PC_61_BM content. Hence, higher PC_61_BM content enhances the charge carrier’s generation in the VTTBNc blends film. This elucidates the reason for the increase in the responsivity and detectivity values with the amount of PC_61_BM ratios. Whereby the determination of both responsivity and detectivity is directly proportional to the number of photocurrent density. The responsivity of VTTBNc:PC_61_BM are measured as 3.40 mA/W, 6.41 mA/W, 10.89 mA/W of ratios 1:0.5, 1:1.0, 1:1:5 at −1 V bias. The detectivities of VTTBNc:PC_61_BM are observed as 1.84 × 10^8^ Jones, 7.26 × 10^8^ Jones, 1.21 × 10^9^ Jones of ratios 1:0.5, 1:1.0, 1:1:5 at −1 V bias. The VTTBNc:PC_61_BM with ratio 1:1.5 is chosen as the optimized blend ratio of VTTBNc:PC_61_BM blends as it has the highest responsivity and detectivity values among the other blend ratios of VTTBNc:PC_61_BM devices.

Whereas for the VTTBNc:PC_71_BM blend, the ratio of 1:1.0 showed the highest detectivity of 3.85 × 10^11^ Jones and 2.31 × 10^9^ Jones at 0 V and −1 V, respectively. On the other hand, ratio 1:1.5 exhibits lower detectivity of 8.48 × 10^10^ Jones and 2.05 × 10^9^ Jones despite having higher responsivity of 9.55 mA/W and 28.48 mA/W at 0 V and −1 V, respectively. The high detectivity of ratio 1:1.0 may be due to the low dark current density obtained. A low dark current is favorable in photodetection applications as it indicates low noise and high detectivity in the OPD device as high dark current will lead to higher device power consumption and may accommodate the photodiode to hardly detect weak signal for signal readout [10]. Thus, the device with a ratio of 1:1.0 was considered an optimized device in VTTBNc:PC_71_BM as it has the highest detectivity values.

Between the two blended devices, VTTBNc:PC_71_BM has the highest responsivity and detectivity values compared to VTTBNc:PC_61_BM. As seen in Table 2, the responsivity measured for optimized ratio of VTTBNc:PC_71_BM (1:1.0) and VTTBNc:PC_61_BM (1:1.5) are (6.96/3.63) mA/W, (15.04/7.38) mA/W and (26.11/10.89) mA/W at 0 V, −0.5 V and −1 V, respectively. In addition to that, the detectivity for both optimized devices are (3.85 × 10^11^/4.41 × 10^9^) Jones, (2.37 × 10^9^/1.53 × 10^9^) Jones and (2.31 × 10^9^/1.21 × 10^9^) Jones at 0 V, −0.5 V and −1 V, respectively. Figure 8 depicts the trend in detectivity and responsivity of the optimized ratio of both blends with respect to bias. According to detectivity values obtained in Table 1 and Table 2, it is evident that the addition of VTTBNc has greatly improved the detectivity limit in VTTBNc:PC_71_BM device to about 186% with respect to PC_71_BM-only device compared to VTTBNc:PC_61_BM device, which increases only around 19% with respect to the PC_61_BM-only device.

Figure 9 depicts the photo-response of VTTBNc:PC_61_BM and VTTBNc:PC_71_BM measured at −1 V in dark (OFF) and light illumination (ON) conditions by measuring the response and recovery time of the OPD device. The response time of the OPD device is counted as the photocurrents jump from the OFF to the ON states, while recovery time is counted from ON to the OFF states. The OPD devices were repeatedly measured in the ON/OFF states in five cycles with a time gap of 5 s, as illustrated in Figure 9a,b. According to Table 3, the measured response and recovery time of VTTBNc:PC_71_BM (1:1.0) ratio are 241 ms and 310 ms, respectively, while for VTTBNc:PC_61_BM (1:1.5), the response and recovery time are recorded as 516 ms and 481 ms, respectively. The results show that the VTTBNc:PC_71_BM (1:1.0) has a smaller or rapid response and recovery time compared to VTTBNc:PC_61_BM (1:1.5). Besides, as seen in Figure 9a, there are spikes or noise observed in VTTBNc:PC_61_BM (1:1 and 1:1.5) at the dark states while no photo-response is detected in 1:0.5 due to the weak signal and high noise detected from the experiment. These spikes might be due to the contribution of high dark current and low photocurrent generation. Meanwhile, high detectivity in VTTBNc:PC_71_BM caused the photo-response to have a stable detection in dark/light state. The main contribution of high dark current in small molecule OPD is due to the carrier injection phenomenon from electrodes that are caused by the low band gap characteristic in small molecules [34].

### 3.4. Photoluminescence and Raman Spectra

Furthermore, the photoluminescence behavior of the optimized ratio for both blends was observed under an excitation wavelength of 325 nm. Figure 10a shows the PL spectra of the optimized ratio of the blends; VTTBNc:PC_61_BM (1:1.5) and VTTBNc:PC_71_BM (1:1.0). Generally, PL measures the radiative illumination of a material. The intensity of PL spectra provides information on the charge carrier’s recombination rate in order to obtain an effective charge transfer within the blended film [35]. It is evident that VTTBNc:PC_71_BM blend film has lower PL intensity than VTTBNc:PC_61_BM, especially in the range of 400–700 nm and 930–1150 nm. The PL spectra of VTTBNc:PC_71_BM are quenched around 22% at 600 nm compared to VTTBNc:PC_61_BM. The percentage of the quenching is calculated based on the difference in the PL intensity of both blends with respect to the VTTBNc:PC_61_BM. This quenching generally suggests that VTTBNc:PC_71_BM has more effective photo-induced charge transfer in the blended film and demonstrates a lower recombination rate of the charge carriers [36].

Furthermore, Raman spectroscopy was used to observe the molecular vibration mode in different blended films. In this characterization, the poor and high-performance blend ratio from VTTBNc:PC_71_BM and VTTBNc:PC_61_BM OPDs was observed. Figure 10b shows the Raman spectra of VTTBNc:PC_71_BM (1:0.5 and 1:1.0) and VTTBNc:PC_61_BM (1:0.5 and 1:1.5). The assignments of the vibrational mode recorded in Table 4 were performed based on the literature [37]. It is observed that VTTBNc:PC_71_BM exhibits a sharp peak at 1566 cm^−1^ in both ratios, which refer to C=C stretch mode. Meanwhile, most peaks in VTTBNc:PC_61_BM have lower intensity and broader peaks. The result also shows two sharp peaks in VTTBNc:PC_71_BM for both ratios that refer to C–H bending mode. From the literature, the sharp peak in the Raman spectra is often associated with the crystallinity and ordered structure of the film [35,38]. Therefore, VTTBNc:PC_71_BM films can be assumed to garner more crystalline and ordered structure compared to VTTBNc:PC_61_BM films. The high crystalline structure in VTTBNc:PC_71_BM supports its high OPD performance due to high charge carrier mobilities that are required for efficient charge extraction [39,40]. On the other side, Raman intensity at 1566 cm^−1^ for VTTBNc:PC_71_BM and 1467 cm^−1^ for VTTBNc:PC_61_BM for ratio 1:0.5 is low compared to other ratios (1:1.0 and 1:1.5) due to the lower content of PC_71_BM and PC_61_BM molecules.

### 3.5. Morphological and Surface Roughness Observations

The morphology and surface roughness of VTTBNc:PC_61_BM and VTTBNc:PC_71_BM thin films were investigated using AFM analysis with scan sizes of 5 × 5 μm and 0.5 × 0.5 μm. As shown in Table 5, four blend ratios with poor and optimized performance from VTTBNc:PC_61_BM (1:0.5 and 1:1.5) and VTTBNc:PC_71_BM (1:0.5 and 1:1.0) were chosen. From the AFM images with 0.5 × 0.5μm scan size, the ratio with higher n-type content exhibits higher surface roughness values which are 0.546 nm for VTTBNc:PC_61_BM (1:1.5) and 0.849 nm for VTTBNc:PC_71_BM (1:1.0). Meanwhile, the surface roughness for the low n-type content (1:0.5) is found to be 0.284 nm for VTTBNc:PC_61_BM and 0.461 nm for VTTBNc:PC_71_BM. As seen in Table 5 with a 5 × 5 μm scan size, the clusters are more visible in a blend with higher n-type content. The presence of clusters might increase the surface roughness values. According to L. Benatto et al. (2020), high surface roughness can enhance the donor–acceptor contact area and assist the charge carrier diffusion in the interpenetrating layer [23]. Besides, the n-types used in the blended film also affect the surface roughness values. The key difference between PC_61_BM and PC_71_BM is the axial distance for fullerene moiety. The axial distance for PC_61_BM was reported to be 0.70 nm and for PC_71_BM was 0.80 nm [41]. The axial distance of fullerene moiety is the length of an allotrope of carbon, and this difference in the axial distance results in different surface roughness values. It can be assumed that the larger axial distance in fullerene moiety contributes to the larger cluster size on the surface. Some of the studies also emphasize the importance of the cluster in order to have charge delocalization and consequently reduce the recombination of charge carriers [42,43].

### 3.6. External Quantum Efficiency (EQE)

Figure 11 shows the EQE spectrum for the optimized OPDs for VTTBNc:PC_61_BM (1:1.5) and VTTBNc:PC_71_BM (1:1.0) as a function of wavelength. This characterization describes the photo-conversion capability of the OPD as the EQE is defined as the ratio between the numbers of charge carriers generated to the numbers of incident photons. It is observed that both OPDs exhibit the highest EQE peak at 400 nm, which is 2.74% for VTTBNc:PC_71_BM and 1.03% for VTTBNc:PC_61_BM. The low percentage of EQE may be due to the low charge carrier generated by the photoactive layer. The low charge carrier generated explained the low photocurrent density measured of both blended devices. The lower EQE values in the NIR region may be due to low light absorption, which lowers the charge carrier generation in the NIR region compared to the visible region.

## 4. Conclusions

In conclusion, the fabrication of OPDs using VTTBNc as a donor and PC_61_BM and PC_71_BM as acceptors were demonstrated. OPD performances were investigated along with the study on the optical, structural and morphological properties of VTTBNc:PC_71_BM and VTTBNc:PC_61_BM of different blend ratios. From the characterizations, we conclude that the optimized ratio obtained for VTTBNc:PC_71_BM and VTTBNc:PC_61_BM blend film is 1:1.0 and 1:1.5, respectively. VTTBNc:PC_71_BM and VTTBNc:PC_61_BM obtained detectivity of 2.31 × 10^9^ Jones and 1.21 × 10^9^ Jones with responsivity values of 26.11 mA/W and 10.89 mA/W, respectively. Both optimized blends obtained acceptable response and recovery time in milliseconds. The response and recovery time in optimized VTTBNc:PC_71_BM were found to be 241 ms and 310 ms, while in optimized VTTBNc:PC_61_BM is 516 ms and 481 ms, respectively. Comprehensively, the VTTBNc:PC_71_BM (1:1.0) has the highest performance among other OPD devices as it has considerably high photocurrent generation compared to other studied devices.

## Figures and Tables

**Figure 1 micromachines-12-01383-f001:**
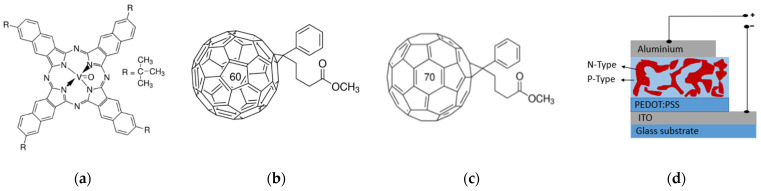
The molecular structure of (**a**) vanadyl 2,11,20,29-tetra tert-butyl 2,3 naphthalocyanine (VTTBNc), (**b**) [6, 6]phenyl-C61-butyric acid methyl ester (PC_61_BM), (**c**) [6, 6]phenyl-C71-butyric acid methyl ester (PC_71_BM) and (**d**) a schematic diagram of organic photodiode.

**Figure 2 micromachines-12-01383-f002:**
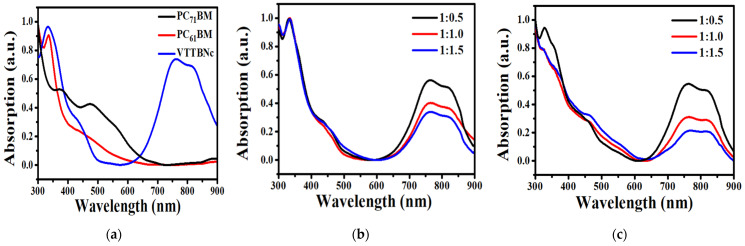
(**a**) Absorption spectra of VTTBNc, PC_61_BM and PC_71_BM thin films. Absorption spectra of (**b**) VTTBNc:PC_61_BM and (**c**) VTTBNc:PC_71_BM of different volumetric ratios.

**Figure 3 micromachines-12-01383-f003:**
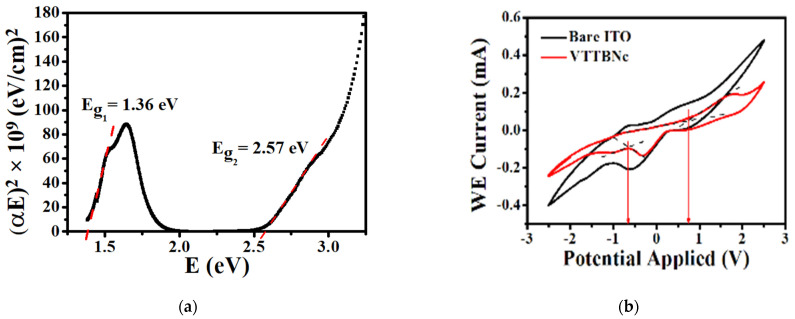
(**a**) Absorption spectrum Tauc plot of VTTBNc thin film. (**b**) Cyclic Voltammogram (CV) of measured current versus applied potential at a scan rate of 75 mV/s.

**Figure 4 micromachines-12-01383-f004:**
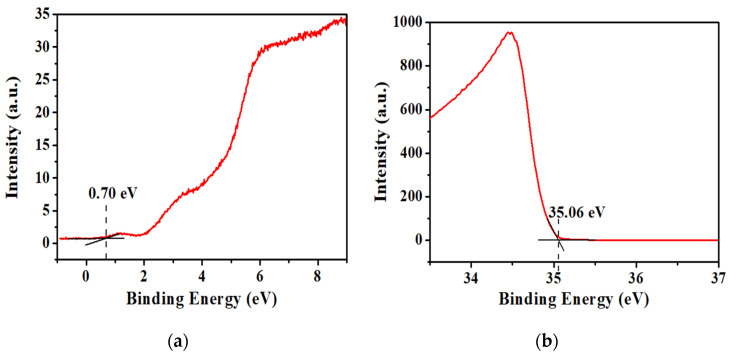
The measured ultra-violet photoelectron spectroscopy (UPS) spectra showing (**a**) highest occupied molecular orbital (HOMO) onset and (**b**) cut-off energy for the determination of VTTBNc’s HOMO level.

**Figure 5 micromachines-12-01383-f005:**
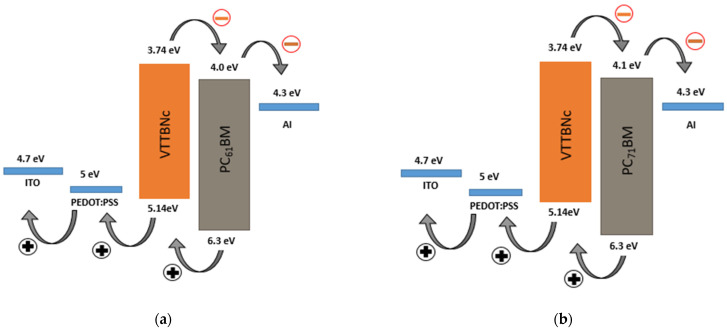
The HOMO-LUMO energy levels and work function for the organic photodiode (OPD) devices (**a**) ITO/PEDOT:PSS/VTTBNc:PC_61_BM/Al (**b**) ITO/PEDOT:PSS/VTTBNc:PC_71_BM/Al.

**Figure 6 micromachines-12-01383-f006:**
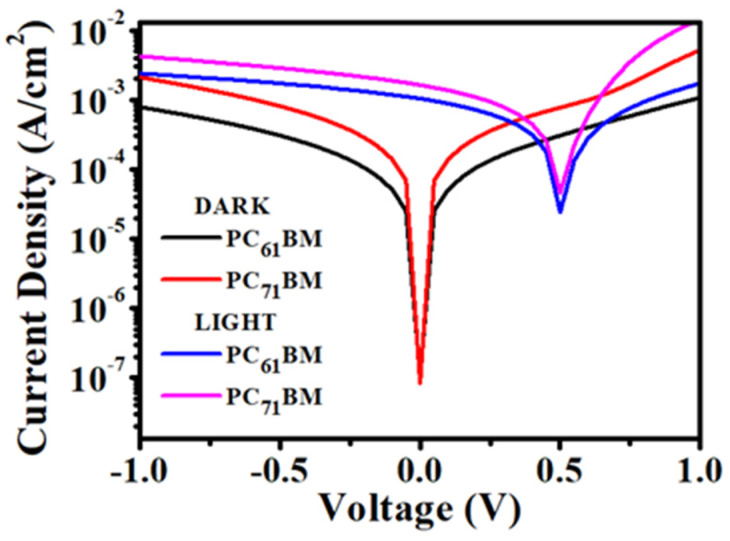
Current density–voltage (J–V) characteristics of PC_61_BM and PC_71_BM under dark and light illumination.

**Figure 7 micromachines-12-01383-f007:**
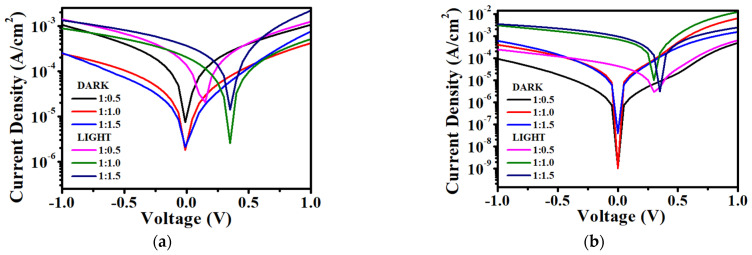
J–V characteristics of (**a**) VTTBNc:PC_61_BM and (**b**) VTTBNc:PC_71_BM of different ratios. J–V characteristics of (**c**) VTTBNc:PC_61_BM and VTTBNc:PC_71_BM of the optimized ratios.

**Figure 8 micromachines-12-01383-f008:**
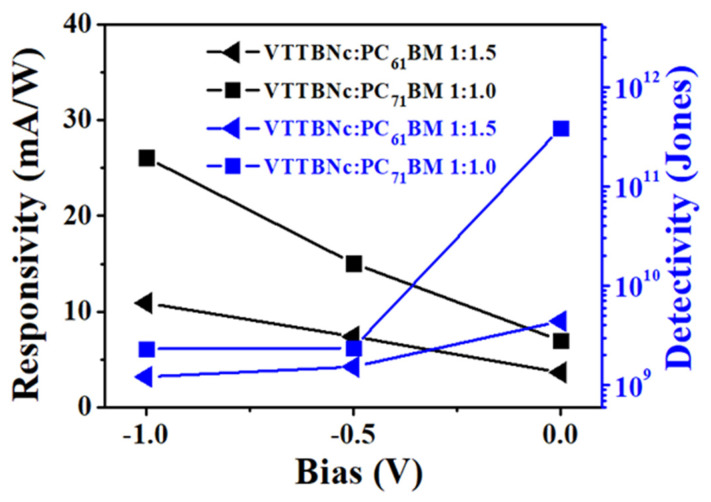
Responsivity and Detectivity of the VTTBNc:PC_61_BM and VTTBNc:PC_71_BM based on the optimized ratios.

**Figure 9 micromachines-12-01383-f009:**
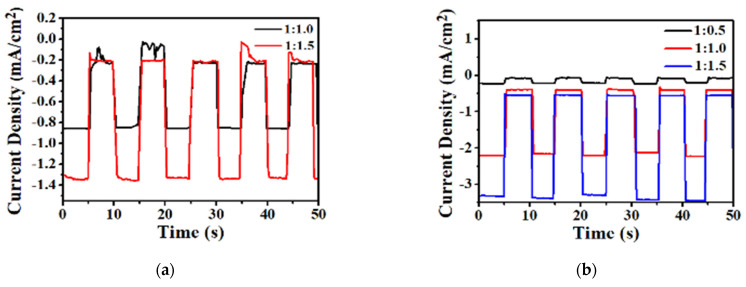
Photo-response of (**a**) VTTBNc:PC_61_BM blends, (**b**) VTTBNc:PC_71_BM blends, (**c**) optimized VTTBNc:PC_61_BM (1:1.5) (**d**) and optimized ratio VTTBNc:PC_71_BM (1:1.0).

**Figure 10 micromachines-12-01383-f010:**
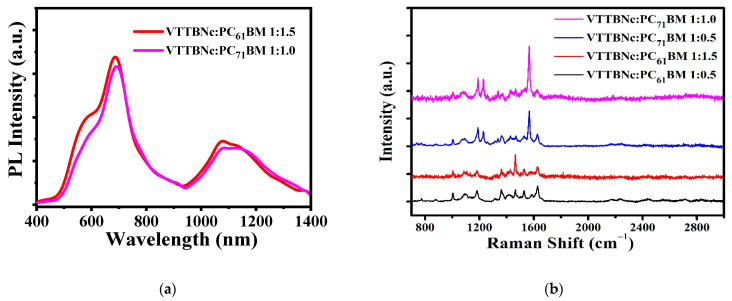
(**a**) Photoluminescence spectra of VTTBNc:PC_61_BM 1:1.5 and VTTBNc:PC_71_BM 1:1.0. (**b**) Raman spectra of VTTBNc:PC_61_BM and VTTBNc:PC_71_BM.

**Figure 11 micromachines-12-01383-f011:**
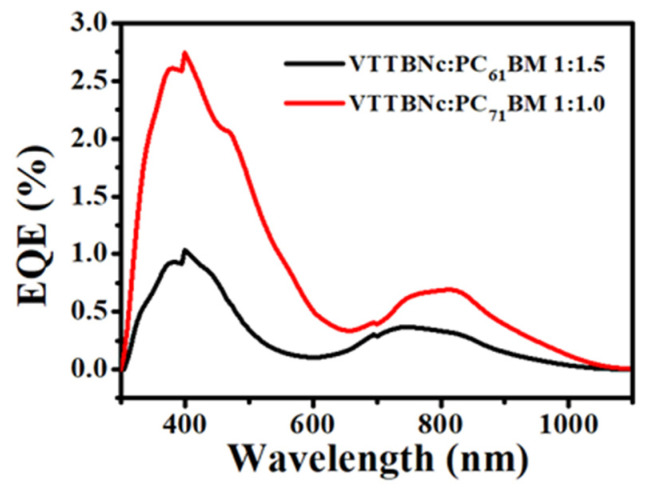
External quantum efficiency (EQE) as a function of wavelength for VTTBNc:PC_61_BM 1:1.5 and VTTBNc:PC_71_BM 1:1.0.

**Table 1 micromachines-12-01383-t001:** Performance of PC_61_BM and PC_71_BM as a photodiode.

Sample	Bias (V)	*J*_D_ (A/cm^2^)	*J*_ph_ (A/cm^2^)	*R* (mA/W)	*D** (Jones)	V_OC_ (V)
PC_61_BM	0	1.04 × 10^−7^	1.05 × 10^−3^	10.50	5.75 × 10^10^	0.5
	−0.5	3.08 × 10^−4^	1.74 × 10^−3^	14.32	1.44 × 10^9^	
	−1.0	7.89 × 10^−4^	2.41 × 10^−03^	16.21	1.02 × 10^9^	
PC_71_BM	0	8.42 × 10^−8^	1.64 × 10^−3^	16.40	9.98 × 10^10^	0.5
	−0.5	8.14 × 10^−4^	2.88 × 10^−3^	20.66	1.28 × 10^9^	
	−1.0	2.15 × 10^−3^	4.27 × 10^−3^	21.20	8.08 × 10^8^	

(*J*_D_ = Dark current density, *J*_ph_ = Photocurrent density, *R* = Responsivity, *D** = Detectivity and V_OC_ = Open-circuit voltage).

**Table 2 micromachines-12-01383-t002:** Performance of VTTBNc:PC_61_BM and VTTBNc:PC_71_BM as a photodiode.

Sample	Ratio	Bias (V)	*J*_D_ (A/cm^2^)	*J*_ph_ (A/cm^2^)	*R* (mA/W)	*D** (Jones)	V_OC_ (V)
VTTBNc:PC_61_BM	1:0.5	0 V	7.48 × 10^−6^	1.35 × 10^−4^	1.28	8.24 × 10^8^	0.15
		−0.5 V	4.13 × 10^−4^	6.75 × 10^−4^	2.62	2.28 × 10^8^	
		−1.0 V	1.06 × 10^−3^	1.40 × 10^−3^	3.40	1.84 × 10^8^	
	1:1.0	0 V	1.82 × 10^−6^	2.03 × 10^−4^	2.01	2.64 × 10^9^	0.35
		−0.5 V	1.07 × 10^−4^	5.17 × 10^−4^	4.10	7.00 × 10^8^	
		−1.0 V	2.43 × 10^−4^	8.84 × 10^−4^	6.41	7.26 × 10^8^	
	1:1.5	0 V	2.11 × 10^−6^	3.65 × 10^−4^	3.63	4.41 × 10^9^	0.35
		−0.5 V	7.27 × 10^−5^	8.11 × 10^−4^	7.38	1.53 × 10^9^	
		−1.0 V	2.51 × 10^−4^	1.34 × 10^−3^	10.89	1.21 × 10^9^	
VTTBNc:PC_71_BM	1:0.5	0 V	1.08 × 10^−9^	4.35 × 10^−5^	0.43	2.34 × 10^10^	0.3
		−0.5 V	1.81 × 10^−5^	1.18 × 10^−4^	1.00	4.15 × 10^8^	
		−1.0 V	9.25 × 10^−5^	2.43 × 10^−4^	1.51	2.76 × 10^8^	
	1:1.0	0 V	1.02 × 10^−9^	6.96 × 10^−4^	6.96	3.85 × 10^11^	0.3
		−0.5 V	1.26 × 10^−4^	1.63 × 10^−3^	15.04	2.37 × 10^9^	
		−1.0 V	3.99 × 10^−4^	3.01 × 10^−3^	26.11	2.31 × 10^9^	
	1:1.5	0 V	3.96 × 10^−8^	9.55 × 10^−4^	9.55	8.48 × 10^10^	0.35
		−0.5 V	1.46 × 10^−4^	2.20 × 10^−3^	20.54	3.00 × 10^9^	
		−1.0 V	6.02 × 10^−4^	3.45 × 10^−3^	28.48	2.05 × 10^9^	

(*J*_D_ = Dark current density, *J*_ph_ = Photocurrent density, *R* = Responsivity, *D**= Detectivity and V_OC_ = Open-circuit voltage).

**Table 3 micromachines-12-01383-t003:** Response and recovery time of the OPDs measured at −1 V bias.

Sample	Ratio	Recovery Time (ms)	Response Time (ms)
VTTBNc:PC_61_BM	1:0.5	-	-
	1:1.0	895	518
	1:1.5	481	516
VTTBNc:PC_71_BM	1:0.5	311	242
	1:1.0	310	241
	1:1.5	336	524

**Table 4 micromachines-12-01383-t004:** Raman shifts of VTTBNc:PC_71_BM and VTTBNc:PC_61_BM.

Assignments	Raman Shift (cm^−1^)			
	VTTBNc:PC_71_BM		VTTBNc:PC_61_BM	
	1:1.0	1:0.5	1:1.5	1:0.5
Benzene ring breath	1003	1003	1003	1003
Ring stretch	1086	1086	1092	1092
C–H bend	1190, 1227	1190, 1227	1180	1180
Pyrrole stretch	1337	1337	1316	1316
Ring stretch	1363, 1430, 1467	1363, 1430, 1467	1363, 1420, 1467	1363, 1415, 1467
Pyrrole stretch	1534	1529	1530	1530
C=C stretch	1566, 1626	1566, 1629	1582, 1629	1587, 1629

**Table 5 micromachines-12-01383-t005:** Atomic Force Microscopy (AFM) images of VTTBNc:PC_61_BM (1:0.5 and 1:1.5) and VTTBNc:PC_71_BM (1:0.5 and 1:1.0).

Scan Size	5 μm × 5 μm	0.5 μm× 0.5 μm
VTTBNc:PC_61_BM (1:0.5)	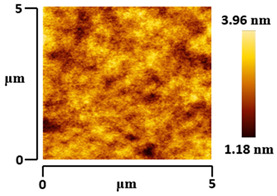	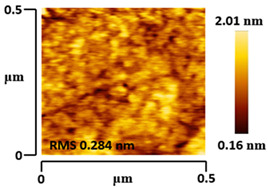
VTTBNc:PC_61_BM (1:1.5)	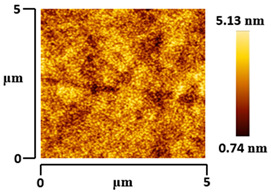	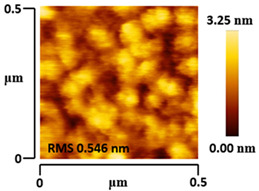
VTTBNc:PC_71_BM (1:0.5)	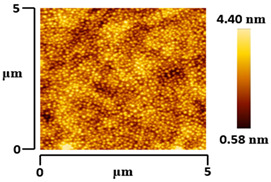	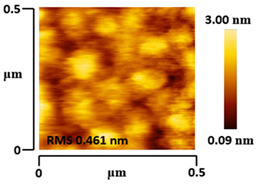
VTTBNc:PC_71_BM (1:1.0)	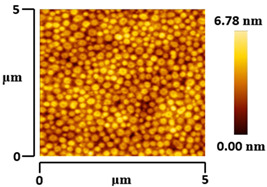	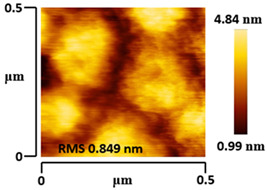

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
