# Peer review of "Naphthalocyanine-Based NIR Organic Photodiode: Understanding the Role of Different Types of Fullerenes"

_micromachines, 2021, doi:10.3390/mi12111383_

Round 1
Reviewer 1 Report
In this manuscript, the authors report a study on the photodetection properties of VTTBNc:PC61BM and VTTBNc:PC71BM bulk heterojunction devices and compare these devices with organic photodiodes made from single layer PC61BM and PC71BM only. The devices were fabricated by solution spin coating and optical absorption, Raman, morphological, electrical characterization had been performed. The manuscript is well structured and the standard of presentation is good. Still, authors need to revise the manuscript by addressing the following points:
Authors should manually add in references to all Figures, Tables and bibliography throughout the manuscript. Readers cannot understand the manuscript without proper referencing.
A novel aspect of this work is the use of VTT-BNc as the donor material for the organic photodiode. In line 54, elaborate on why a vanadium containing molecule is selected and is this related to the narrow HOMO LUMO gap of this donor.
In the Experimental section, lines 74 and 77, state the ambient condition during spin coating.
For Fig. 3a, provide the unit of (alpha E)^2 in the vertical axis and explain why there are evidently two spectral regions of strong absorption. Relate these absorption regions to the electronic structure of the VTTBNc film. A linear fit to the experimental data needs to be added to show the optical energy gap is 1.36 eV. A similar fit can be done for the absorption data above 2.5 eV. Show the value of this second optical energy gap.
In Fig. 3b, indicate the onset potential of oxidation and reduction. In the accompanying text (line 142 - 148), split equation 1 into two equations, one for the energy of the HOMO and another one for the energy of the LUMO.
A similar problem of presentation occurs in section 3.3 (Line 187-188). Discuss the responsivity of PC71BM and PC61MB single layer devices separately. Why are charge carriers thermally generated at the contacts in a PC71BM device when a voltage is applied?
We recommend Table 1 (and indeed Table 2) to be re-organized so that the columns are device properties and the rows are the two single layer (or BHJ) devices at different biases.
In line 197, explain a bit more on the properties of the VTTBNc: PC61BM devices before moving on to the VTTBNc:PC71BM bulk heterojunction devices.
In line 217, authors need to mention that Fig. 7 shows the trend of detectivity and responsivity with respect to bias. For the caption, please mention that the blends are at their respective optimized ratios.
In line 262-263, the authors wrote that the PL of VTTBNc:PC71BM is quenched 20% with respect to VTTBNc:PC61BM. Explain how the 20% is deduced from the PL data.
In section 3.5, clarify what is the axial distance for the fullerene moiety. For bulk heterojunction devices, the basic topographic image is not informative. Have the authors considered the use of phase contrast (non-contact) AFM imaging mode which is more sensitive to changes in the chemical composition and hence may show the bulk heterojunction morphology more clearly? This data is possibly very useful to explain the low EQE reported in section 3.6. This paragraph needs to be expanded to explain why the EQE is so low. What are the main losses in the device? Authors should include an estimate of the internal quantum efficiency to show whether the optical losses from reflection is significant or not.
Author Response
Response to Reviewer 1 Comments:
Manuscript ID: micromachines-1428359
Title: Naphthalocyanine-based NIR Organic Photodiode: Understanding the Role of Different Types of Fullerenes
We are much thankful to the editor and reviewer for their deep and thorough review. We have revised the suggested research paper in the light of their useful suggestions and comments. Some figures have been altered/ added to meet the suggestions. We hope our revision have improved the paper to a level of satisfaction. Answer to their specific comment/ suggestions/queries are as follows.
Reviewer 1:
In this manuscript, the authors report a study on the photodetection properties of VTTBNc:PC61BM and VTTBNc:PC71BM bulk heterojunction devices and compare these devices with organic photodiodes made from single layer PC61BM and PC71BM only. The devices were fabricated by solution spin coating and optical absorption, Raman, morphological, electrical characterization had been performed. The manuscript is well structured and the standard of presentation is good. Still, authors need to revise the manuscript by addressing the following points:
Authors should manually add in references to all Figures, Tables and bibliography throughout the manuscript. Readers cannot understand the manuscript without proper referencing.
Justification / Action:
- The references to all Figures, Tables and bibliography throughout the manuscript has already been added.
A novel aspect of this work is the use of VTTBNc as the donor material for the organic photodiode. In line 54, elaborate on why a vanadium containing molecule is selected and is this related to the narrow HOMO LUMO gap of this donor.
Justification / Action:
- The use of VTTBNc as a donor in the structure is due to the ability of this material to absorb light in the NIR region and its low band gap property. Generally, the low band gap of donor offers smaller energy offset between HOMO or LUMO of donor and acceptor and consequently minimizes the loss due to the charge transfer state between donor-acceptor [19].
[19] Cheng, P. and Y. Yang, Narrowing the band gap: the key to high-performance organic photovoltaics. Accounts of chemical research, 2020. 53(6): p. 1218-1228
In the Experimental section, lines 74 and 77, state the ambient condition during spin coating.
Justification / Action:
- The ambient condition during the spin coating process of PEDOT:PSS has been stated.
“Next, a PEDOT:PSS solution (PH1000 from H.C Stack) was filtered using 0.45 μm nylon filter and spin coated in the room temperature at 3000 rpm for 60 s on top of cleaned ITO substrate to form a 40 nm thin layer and annealed for 30 min at 130˚C as the hole transport layer.”
For Fig. 3a, provide the unit of (alpha E)^2 in the vertical axis and explain why there are evidently two spectral regions of strong absorption. Relate these absorption regions to the electronic structure of the VTTBNc film. A linear fit to the experimental data needs to be added to show the optical energy gap is 1.36 eV. A similar fit can be done for the absorption data above 2.5 eV. Show the value of this second optical energy gap.
Justification / Action:
- The unit of (alpha E)^2 in the vertical axis has been added in the Figure 3(a).
- The explanation of two spectral regions of strong absorption of VTTBNc has been added and relation between absorption and electronic structure of VTTBNc has been further mentioned in section 3.2.
“The absorption peak at the B-band correlates to the excitation of localized π-π* transition that occurred between the bonding and anti-bonding molecules. Meanwhile, the absorption peak at the Q-band is related to the charge transfer within the core unit [20, 21].”
- The linear fit together with the values of the first and the second optical band gap has been added in Figure 3(a).
[20] Dhanya, I. and C. Menon, Surface morphological, structural, electrical and optical properties of annealed vanadyl tetra tert-butyl 2, 3 naphthalocyanine thin films. Vacuum, 2012. 86(9): p. 1289-1295.
[21] Namepetra, A., et al., Understanding the morphology of solution processed fullerene-free small molecule bulk heterojunction blends. Physical Chemistry Chemical Physics, 2016. 18(18): p. 12476-12485.
In Fig. 3b, indicate the onset potential of oxidation and reduction. In the accompanying text (line 142 - 148), split equation 1 into two equations, one for the energy of the HOMO and another one for the energy of the LUMO.
Justification / Action:
- The onset potential of oxidation and reduction have been indicated in line 174 - 175.
“The oxidation and reduction onset potential of VTTBNc are observed at 0.742 V and -0.662 V, respectively.”
- The equation of HOMO and LUMO has been individual stated.
|
HOMO = -e (EOX + 4.4eV) |
|
or LUMO = -e (ERED + 4.4eV) |
A similar problem of presentation occurs in section 3.3 (Line 187-188). Discuss the responsivity of PC71BM and PC61MB single layer devices separately. Why are charge carriers thermally generated at the contacts in a PC71BM device when a voltage is applied?
Justification / Action:
- The responsivity of PC71BM and PC61BM has been discussed separately.
“The responsivity of PC71BM is measured as 16.40 mA/W, 20.66 mA/W, 21.20 mA/W at 0 V, -0.5 V, -1 V bias, respectively. While the responsivity of PC61BM is measured as 10.50 mA/W, 14.32 mA/W, 16.21 mA/W at 0 V, -0.5 V, -1 V bias, respectively, as listed in Table 1.”
- In the dark state, both devices show the presence of the dark current that might be influence by thermally generation of the charge carriers. The explanation of high thermal generation of charge carriers at the contacts in PC71BM has been added as follows:
“As PC71BM obtained stronger absorption and higher photocurrent generation, it also contains a higher charge carrier generation compared to PC61BM, and in turn, tends to have a high thermally generation of charge carriers from the two electrodes contact.”
We recommend Table 1 (and indeed Table 2) to be reorganized so that the columns are device properties and the rows are the two single layer (or BHJ) devices at different biases.
Justification / Action:
- Table 1 and table 2 has been reorganized.
In line 197, explain a bit more on the properties of the VTTBNc: PC61BM devices before moving on to the VTTBNc:PC71BM bulk heterojunction devices.
Justification / Action:
- The explanation of VTTBNc:PC61BM has been added as follow:
“For the VTTBNc:PC61BM devices, the result shows the increase in the photocurrent density with increasing PC61BM content. Hence, higher PC61BM content enhances the charge carrier’s generation in the VTTBNc blends film. This elucidates the reason for the increase in the responsivity and detectivity values with the amount of PC61BM ratios. Whereby the determination of both responsivity and detectivity is directly proportional to the number of photocurrent density. The responsivity of VTTBNc:PC61BM are measured as 3.40 mA/W, 6.41 mA/W , 10.89 mA/W of ratios 1:0.5, 1:1.0, 1:1:5 at -1V bias. The detectivity of VTTBNc:PC61BM are observed as 1.84x108 Jones, 7.26x108 Jones, 1.21x109 Jones of ratios 1:0.5, 1:1.0, 1:1:5 at -1V bias. The VTTBNc:PC61BM with ratio 1:1.5 is chosen as the optimized blend ratio of VTTBNc:PC61BM blends as it has the highest responsivity and detectivity values among the other blend ratios of VTTBNc:PC61BM devices.”
In line 217, authors need to mention that Fig. 7 shows the trend of detectivity and responsivity with respect to bias. For the caption, please mention that the blends are at their respective optimized ratios.
Justification / Action:
- The trend of detectivity and responsivity with respect to bias of Fig. 7 (currently known as figure 8) has been mentioned in line 263-264.
- The caption whereas the blends are at their respective optimized ratios has been added in the caption of figure 7 (currently known as figure 8) .
In line 262-263, the authors wrote that the PL of VTTBNc:PC71BM is quenched 20% with respect to VTTBNc:PC61BM. Explain how the 20% is deduced from the PL data.
Justification / Action:
- The explanation of how 22% is deduced from the PL data has been described in line 308-310.
“The PL spectra of VTTBNc:PC71BM is quenched around 22% at 600 nm compared to VTTBNc:PC61BM. The percentage of the quenching is calculated based on the difference on the PL intensity of both blends with respect to the VTTBNc:PC61BM.”
In section 3.5, clarify what is the axial distance for the fullerene moiety. For bulk heterojunction devices, the basic topographic image is not informative. Have the authors considered the use of phase contrast (non-contact) AFM imaging mode which is more sensitive to changes in the chemical composition and hence may show the bulk heterojunction morphology more clearly? This data is possibly very useful to explain the low EQE reported in section 3.6. This paragraph needs to be expanded to explain why the EQE is so low. What are the main losses in the device? Authors should include an estimate of the internal quantum efficiency to show whether the optical losses from reflection is significant or not.
Justification / Action:
- The meaning of the axial distance for the fullerene moiety has been clarified.
“The axial distance of fullerene moiety is the length of an allotrope of carbon.”
- Thank you so much for suggesting us to consider using phase contrast AFM imaging mode and IQE, in order for us to gain for information on the studies. Unfortunately, at this stage we did not include these two components in our current studies. We definitely will include the phase contrast AFM imaging mode and IQE data in our future studies.

Reviewer 2 Report
Attached file

Author Response
Response to Reviewer 2 Comments:
Manuscript ID: micromachines-1428359
Title: Naphthalocyanine-based NIR Organic Photodiode: Understanding the Role of Different Types of Fullerenes
We are much thankful to the editor and reviewer for their deep and thorough review. We have revised the suggested research paper in the light of their useful suggestions and comments. Some figures have been altered/ added to meet the suggestions. We hope our revision have improved the paper to a level of satisfaction. Answer to their specific comment/ suggestions/queries are as follows.
Reviewer 2:
The introduction section is very short and contain minimum information regarding this work and motivation. The author should explain the importance of this material in details with some references.
Justification / Action:
- More information regarding this work and motivation have been added in the Introduction part with the references.
In line 38-39 authors mentioned the key parameters of OPD such as detectivity, responsivity and external quantum efficiency (EQE). In line 48-49 authors focus is on sensitivity of OPD. Clarify in the last portion of introduction section which of these parameters are improved in your device with VTTBNc layer.
Justification / Action:
- The sensitivity of OPD is refer to the responsivity and detectivity values. To avoid the confusion of this statement, the sensitivity of OPD has been edited to the ‘high performance OPD’ in line 48.
In line 132-133 authors mentioned that the bandgap achieved is slightly low as compared to cited ref. The author should clarify here weather the achieved results are good or not.
Justification / Action:
- The explanation of the slight difference of the band gap has been added.
“The slight difference in the band gap value is due to the red-shifted absorption range observed in this experiment, resulting in lower band gap values. Besides, the red-shifted absorption is good in enhancing the OPD’s detection to the longer wavelength.”
The author mentioned that the detectivity of their material is d 2.31x109 Jones which is very less as compared with detectivity reported by by Zheng et al DOI: 10.1039/c8tc00437d> how this material will be consider better then the previous report?
Justification / Action:
- The previous report by Zheng et al, used conjugated polymer (PBBTPD) as the donor materials and fabricated based on the solution-processed. In this study, we are using VTTBNc which belong to the small molecule group. Therefore, we added some of advantages of using small molecules compared to conjugated polymer in the introduction.
“Small molecules are an organic compound that possesses defined molecular structure with low molecular weight. They are free from batch-to-batch variation and consequently improve the fabrication repeatability compared to the conjugated polymer. Dong et al. emphasized that the improvement of fabrication repeatability can lead to a greater tendency to form ordered domains and provide higher charge carrier mobilities [17]. On top of that, small molecules evade unwanted features of macromolecules like chain twists and chain-end defects that give rise to structural disorder and low-lying trap states [18]. It is also much cheaper than conjugated polymers and thus making it a reasonable material to develop and study.”
[17] Dong, H., et al., Organic photoresponse materials and devices. Chemical Society Reviews, 2012. 41(5): p. 1754-1808.
[18] Collins, S.D., et al., Small is powerful: recent progress in solution‐processed small molecule solar cells. Advanced Energy Materials, 2017. 7(10): p. 1602242.
In line 195 author mentioned that the result shows that VTTBNc:PC 61 BM (1:1.5) has the highest responsivity compared with other ratios. While the author mentioned in line 200 that these results may be due to low dark current. Why may be? are you not sure about the results what other factors can affect the results?. more explanation is required.
Justification / Action:
- The highest responsivity of ratio 1:1.5 is actually related to the performance of VTTBNc:PC61BM. Meanwhile, the explanation of “The high detectivity of ratio 1:1.0 may be due to the low dark current density obtained.” is related to the VTTBNc:PC71BM To avoid misunderstanding in reading this manuscript, we have divided the explanation of VTTBNc:PC61BM and VTTBNc:PC71BM into 2 paragraphs.
In line 206 authors mentioned VTTBNc:PC 71 BM seems to ? what do u mean by seems to have? This confuses me. Author should be sure about the results. Kindly check such words like may be, seems to be etc throughout the manuscript.
Justification / Action:
- Proper words have been used throughout the manuscript.
There are many grammatical errors, recheck the manuscript once again for all typo errors.
Justification / Action:
- All the grammatical errors have been corrected and checked.

Reviewer 3 Report
The authors submitted a report on naphthalocyanine-based near-infrared (NIR) organic photodiode. First of all, I need to mention that it is quite a long report trying to tell more than a single story. As a result, there are too many highlights, and it is more like a Christmas tree. The reader does not know which highlight to focus on. The reduction of the content can enormously improve the manuscript quality. It should be better to say less and understand more.
Nevertheless, the main problem of the manuscript is not the storytelling but the scientific quality and data credibility. If the experimental methods are not explained well, everything is doubtful, and the experimental evidence does not support the conclusions. Several examples:
- electrochemical methods: Authors are using "using 0.01M hydroquinone in HCl with pH 7". Since hydrochloric acid (HCl) has a pH range from 1.5 to 3.5; it is probably the aqueous solution. The authors used electrochemical cyclic voltammetry to estimate the redox potentials (and evaluate the HOMO-LUMO levels). However, the measurement of redox potentials in aqueous solutions should be in the range of approx. -1 to +1 V since above the potential window, the water dissociation takes place, and the results are not reliable. Therefore, authors should use different solvents (such as acetonitrile) to have a larger potential window if required. Furthermore, the explanation on energy level estimation is incorrect. The potential of standard hydrogen electrode (SHE) in respect to vacuum is 4.6 eV; however, the 0.2 eV is an Ag/AgCl shift in respect to the SHE. As a result, the potential of 4.4 eV represents the absolute potential of Ag/AgCl electrodes in respect to vacuum. In addition, the estimation of redox peaks in Fig. 3(b) is unclear and extremely doubtful.
- optical methods: The authors applied the Tauc plot to estimate the optical energy gap. The estimated value is significantly lower than the expected (reported) one. The authors discuss the reason; however, the most probably is the result evaluation incorrect. The Tauc plot may also have a different power exponent depending on the nature of transitions (allowed/forbidden, direct/indirect). Hence, the estimated optical energy gap may differ a lot depending on the transition.
- photoresponse measurement: the nature of the photoresponse should be estimated well. Authors are scientists; hence, the behaviour must be investigated in detail. For example, is it exponential or power-law (charge diffusion)? What does the response/recovery time mean, and how it was estimated? Here I need to mention that when I took Authors' data (got from the Figure), the response time period was very different from reported values. Again, as a reader, I need to doubt on the results and discussion.
In conclusion, the present form of the manuscript raises more questions than it is common, and I need to require major changes before any discussion on the manuscript acceptance. On the other hand, I strongly recommend that the authors make major changes and additional experiments and submit the manuscript once again.
Author Response
Response to Reviewer 3 Comments:
Manuscript ID: micromachines-1428359
Title: Naphthalocyanine-based NIR Organic Photodiode: Understanding the Role of Different Types of Fullerenes
We are much thankful to the editor and reviewer for their deep and thorough review. We have revised the suggested research paper in the light of their useful suggestions and comments. Some figures have been altered/ added to meet the suggestions. We hope our revision have improved the paper to a level of satisfaction. Answer to their specific comment/ suggestions/queries are as follows.
Reviewer 3:
The authors submitted a report on naphthalocyanine-based near-infrared (NIR) organic photodiode. First of all, I need to mention that it is quite a long report trying to tell more than a single story. As a result, there are too many highlights, and it is more like a Christmas tree. The reader does not know which highlight to focus on. The reduction of the content can enormously improve the manuscript quality. It should be better to say less and understand more.
Nevertheless, the main problem of the manuscript is not the storytelling but the scientific quality and data credibility. If the experimental methods are not explained well, everything is doubtful, and the experimental evidence does not support the conclusions. Several examples:
electrochemical methods: Authors are using "using 0.01M hydroquinone in HCl with pH 7". Since hydrochloric acid (HCl) has a pH range from 1.5 to 3.5; it is probably the aqueous solution. The authors used electrochemical cyclic voltammetry to estimate the redox potentials (and evaluate the HOMO-LUMO levels). However, the measurement of redox potentials in aqueous solutions should be in the range of approx. -1 to +1 V since above the potential window, the water dissociation takes place, and the results are not reliable. Therefore, authors should use different solvents (such as acetonitrile) to have a larger potential window if required. Furthermore, the explanation on energy level estimation is incorrect. The potential of standard hydrogen electrode (SHE) in respect to vacuum is 4.6 eV; however, the 0.2 eV is an Ag/AgCl shift in respect to the SHE. As a result, the potential of 4.4 eV represents the absolute potential of Ag/AgCl electrodes in respect to vacuum. In addition, the estimation of redox peaks in Fig. 3(b) is unclear and extremely doubtful.
Justification / Action:
- The pH of HCl in this work is 3.0 and the info has been corrected in the text, line 104. The explanation on the energy level estimation has been added in manuscript. The ultraviolet photoelectron spectroscopy (UPS) result has been added to support the measurement of energy levels obtained from C The sentence related to the 4.4eV as the absolute potential of Ag/AgCl electrode relative to the vacuum level has been included within the manuscript.
optical methods: The authors applied the Tauc plot to estimate the optical energy gap. The estimated value is significantly lower than the expected (reported) one. The authors discuss the reason; however, the most probably is the result evaluation incorrect. The Tauc plot may also have a different power exponent depending on the nature of transitions (allowed/forbidden, direct/indirect). Hence, the estimated optical energy gap may differ a lot depending on the transition.
Justification / Action:
- The y-axis of the Tauc plot is (αhν)n, where n = 2 (direct allowed transition), n = 1/2 (indirect allowed transition), n = 3/2 (forbidden direct transition) and n = 1/3 (forbidden indirect transition). In this case, the band gap of VTTBNc is determined by direct allowed transition (n = 2). The material is based on the basic absorption process (allowed transition) and the graph in the Tauc plot where y-axis is (αhν)2 shows the linear parts of the graph.
photoresponse measurement: the nature of the photoresponse should be estimated well. Authors are scientists; hence, the behaviour must be investigated in detail. For example, is it exponential or power-law (charge diffusion)? What does the response/recovery time mean, and how it was estimated? Here I need to mention that when I took Authors' data (got from the Figure), the response time period was very different from reported values. Again, as a reader, I need to doubt on the results and discussion.
Justification / Action:
- The photo-response is measured using Keithley 236 Source Measure Unit equipped with Oriel (Xenon arc lamp) solar simulator with illumination of 100 mW/cm2 input power. The measurement was taken place to study the response time and recovery time when the light in the switch ON and OFF states, respectively. Thus, the explanation regarding this has been added in lines 278-282.
“The response time of the OPD device is counted as the photocurrents jump from the OFF to the ON states, while recovery time is counted from ON to the OFF states. The OPD devices were repeatedly measured in the ON/OFF states in five cycles with a time gap of 5 seconds, as illustrated in Figure 8(a) & (b).”
- For the difference in data values, we already double check the values and there is no any difference in the data values. But, to give a better view, we change the row and column in the Table 3, so that it can be understood more.
In conclusion, the present form of the manuscript raises more questions than it is common, and I need to require major changes before any discussion on the manuscript acceptance. On the other hand, I strongly recommend that the authors make major changes and additional experiments and submit the manuscript once again.

Round 2
Reviewer 3 Report
The authors modified the manuscript and fixed several issues raised by the Reviewer. On the other hand, some corrections opened new issues.
The authors corrected the supporting electrolyte pH to the value of 3. Now, I believe that it is hydrochloric acid (HCl). However, I cannot believe that the Authors used indium tin oxide (ITO) working electrode since it is damaged in strong acids. In other words, hydrochloric acid is commonly used to etch the ITO layers. Furthermore, it is hard to believe that the investigated organic materials can resist acid. In addition, the authors did not explain the estimation of redox onsets. The great difference between the onset (approx. 0.8V) and the peak (approx. 1.8 V) gives a possibility of huge error. As a result, Authors must explain their experimental methods in detail. The present state of the manuscript does not offer credible information; hence, I cannot support it.